# Widespread impact-generated porosity in early planetary crusts

Sean E. Wiggins [1] ✉, Brandon C. Johnson [1,2], Gareth S. Collins [3], H. Jay Melosh [1,2,5] & Simone Marchi [4]

NASA's Gravity Recovery and Interior Laboratory (GRAIL) spacecraft revealed the crust of the Moon is highly porous, with ~4% porosity at 20 km deep. The deep lying porosity discovered by GRAIL has been difficult to explain, with most current models only able to explain high porosity near the lunar surface (first few kilometers) or inside complex craters. Using hydrocode routines we simulated fracturing and generation of porosity by large impacts in lunar, martian, and Earth crust. Our simulations indicate impacts that produce 100–1000 km scale basins alone are capable of producing all observed porosity within the lunar crust. Simulations under the higher surface gravity of Mars and Earth suggest basin forming impacts can be a primary source of porosity and fracturing of ancient planetary crusts. Thus, we show that impacts could have supported widespread crustal fluid circulation, with important implications for subsurface habitable environments on early Earth and Mars.

Understanding the origin and evolution of planetary crustal porosity is of particular interest because crustal porosity has a strong effect on thermal, magmatic, and hydrothermal processes occurring early in planetary history[1–5] as well as on the prospect of habitable niches outside of Earth. The Moon provides a unique window into the evolution of ancient planetary crusts because its early porosity has mainly only been affected by subsequent impacts, especially on the lunar farside which has experienced less widespread volcanism than the nearside. NASA's Gravity Recovery and Interior Laboratory (GRAIL) mission provided an unparalleled look into the interior structure of the Moon revealing that the lunar crust is much less dense and therefore more porous than previously thought[6,7]. The uppermost ~4 km of the lunar highland crust has an average crustal porosity of 12%, and this porosity decreases to 4% at depths of 20 km[6,8].

The spatial distribution of porosity in the lunar crust suggests that impacts are the likely source of porosity;[6,9–11] however, it is unclear how impacts produce substantial porosity at depth and outside the crater. Previous impact simulations with dilatancy demonstrated how shear deformation can create pore space throughout the crust inside the crater and in the ejecta blanket, but this process cannot produce significant porosity at depth outside the crater rim crest[12,13]. On the other hand, observations indicate that a large fraction of the porosity generated during impact events is created outside the crater rim crest[6,14]. While the porosity of the upper few kilometers can be explained by ejecta deposition, another mechanism is required to explain porosity found deep within the lunar crust, as the maximum thickness of large-scale basin ejecta is about 5 km closer to the basin's rim[3,14–16].

Here we show that appreciable porosity deep in the lunar crust and outside of crater rim crests is likely the result of in situ tensile fragmentation during impacts. During a hypervelocity impact an initial shockwave propagates into the projectile and target and is followed by a tensile rarefaction wave, or tensile relief wave, that propagates from free surfaces. The succession of a shock and rarefaction sets up the flow that excavates the crater[17]. The rarefaction wave can produce strong tensile stresses and high strain rates, triggering dynamic fragmentation[18]. This dynamically fragmented material represents a large volume within the lunar crust that should have some porosity[19]. The amount of porosity generated by in situ fragmentation, however, has not been determined. Here we examine the creation of deep lying porosity by tensile fragmentation and show that it can explain the porosity observed with GRAIL.

[1]Department of Earth, Atmospheric, and Planetary Sciences, Purdue University, West Lafayette IN 47907, USA. [2]Department of Physics and Astronomy, Purdue University, West Lafayette, IN 47907, USA. [3]Department of Earth Science and Engineering, Imperial College London, London SW7 2AZ, UK. [4]Southwest Research Institute, Boulder, CO 80302, USA. [5]Deceased: H. Jay Melosh. ✉e-mail: wigginss@purdue.edu

## Results and discussion

### Impact-induced porosity on the Moon

To investigate porosity generation by tensile dynamic fragmentation in large impacts we simulated a range of impactor sizes (1–1000 km diameters) and target gravity to mimic the Moon, Mars and Earth, using the multimaterial, multirheology iSALE shock physics code[18,20–23]. Previous work has studied only shear-induced porosity via dilatancy, whereas in our work here, we have included a tensile fragmentation routine[19] and a tensile porosity routine. All of our simulations only simulate through the excavation stage of crater development, as this is when tensile fragmentation occurs and it is before the initial porosity structure is modified by gravitational collapse within the crater, ejecta landing, etc. During the next stage of crater development, the modification stage, dilatancy would dominate near the crater[12].

The tensile porosity created by excess negative pressure is a direct result of the rarefaction wave that trails behind the compressive shock wave. By convention tensile stresses are positive and pressure is defined as the negative of one third the trace of the stress tensor. Thus, when the mean normal stress is compressive the pressure is positive and when the mean normal stress is tensile the pressure is negative[24]. The resulting porosity field is significantly different from that produced by a version of iSALE run without the tensile porosity routine (Fig. 1). Even before the modification stage it is clear that a substantial amount of porosity has been created in the material nearest the impact crater. This is a striking difference from previous simulations where only dilatancy was considered, as in those simulations most porosity was created during the modification stage of crater formation[12]. Another remarkable difference when accounting for tensile porosity is the generation of significant porosity deep beneath the surface, extending tens of kilometers down from the point of initial contact (Fig. 1b). Even with the smallest impactor in our simulations (1 km in diameter), substantial porosity is created both near the surface and deep within the crust, with small amounts of porosity generated tens of kilometers beneath the surface (Fig. 1b). For comparison, this is very different from simulations that only consider shear-induced porosity, in which porosity above 0.1% was only generated down to a depth of ~10 km (Fig. 1a), approximately half the depth of the deepest porosity created in tension. In general, for a 1 km diameter impactor, the tensile porosity routine results in 1–100

times more porosity at depth and in the near surface (Supplementary Fig. 4).

For a larger impactor, 10 km in diameter, we observe a similar distribution of porosity, but on a larger scale due to the increase in impactor size. Interestingly, high porosities of ~1% exists even 250 km radially away, even if within the first 5 km of lunar crust. We also find that porosity exceeding 0.01% extends to depths of ~10 km even at radial distances of 250 km (Fig. 2a). While this porosity is modest, our simulations of impacts into an already porous crust agree with previous work which shows that impacts remove porosity close to the point of impact in an already porous target and will increase porosity further away, especially outside the crater. However, our simulations of impacts into pre-porous target crusts indicate that porosity production is additive deep within the lunar crust to yet to be determined threshold (see Supplemental Material). The cumulative effect of many impacts of this size over the bombardment history of the Moon could have increased the bulk porosity of the crust.

Our largest impactor simulation (i.e. 100 km diameter, Fig. 2b) best exemplifies the important role basin forming impacts have on creating large and deep porosity fields. The porosity generated directly underneath the crater does not extend deep into the Moon due to the high overburden pressure. Overburden pressure is the pressure due to overlying material being acted upon by gravity; this overburden pressure must be overcome by the negative pressure of the rarefaction wave in order to continue fracturing and fragmenting material[16], which is why no tensile porosity is accounted for beneath the crater of the 100 km diameter impact scenario (Fig. 2b). However, significant porosity is generated in the near surface of the target. Even 1000 km from the initial contact site, the tensile stresses in the near surface region of this large-scale impact generate porosity to a depth of ~50 km (Fig. 2b). This porosity structure is consistent with estimates of lunar crustal porosity (i.e. a few percent down to depths around 30 km)[6,8]. The great radial extent of this porosity implies that 100–1000 km scale lunar basins alone are capable of producing all of the observed lunar porosity.

### Impact-induced porosity on other bodies

To understand the production of porosity on other terrestrial planets during basin forming impacts, we model 100 km diameter impactors striking the same target with the same conditions, but with a surface

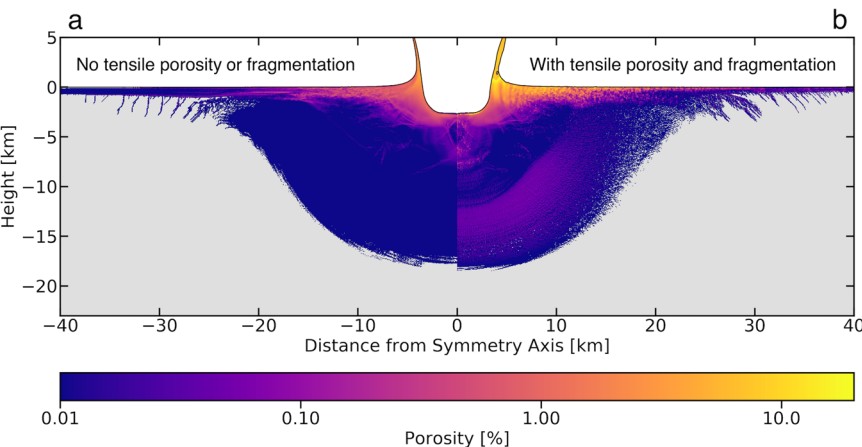

**Fig. 1 | Porosity with and without tensile porosity.** Result of a 1 km diameter impactor into the lunar surface at 15 km/s. The logarithmic color bar on the bottom is applicable to both panels; material with 0% porosity is colored gray. Results without the tensile porosity routine or dynamic fragmentation are on the left (**a**) and those including them are on the right (**b**). White color in the figures in this paper represents void space, or vacuum. Unless otherwise stated the tensile porosity routine and the fragmentation routine are used in all simulations. It is important to note that since these simulations were ended well before the crater is settled, so the porosity distribution around the crater, particularly the floor of the crater, can change substantially. For additional comparison please see Supplementary Text 4 in our supplementary materials.

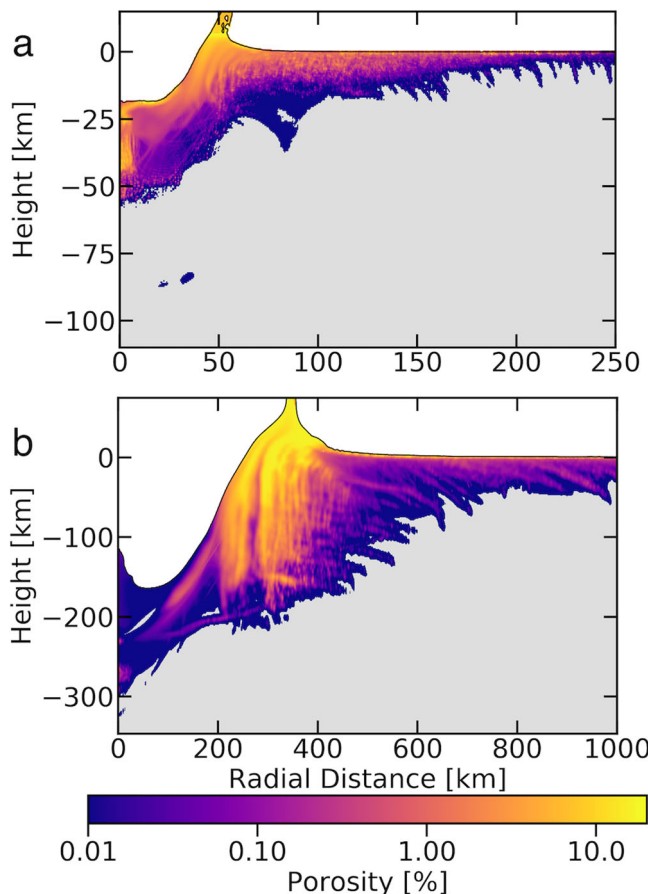

**Fig. 2 | Porosity results for 10 and 100 km diameter impactors. a** Plot of porosity 150 s after initial contact of a 10 km diameter impactor with pristine lunar surface at 15 km/s. **b** Porosity values in an initially intact lunar surface 500 s after a 100 km diameter impactor struck the surface at 15 km/s. The material that now has porosity is colored according to the log color bar on the right. All gray colored material is pristine, meaning it has no porosity. Note both (**a**) and (**b**) are vertically exaggerated, with aspect ratios of 1.359 and 1.6 respectively. In frame (**b**) porosity is produced >1000 km from the point of impact, but this is outside of our high-resolution zone. The low porosity values found within the crater in frame (**b**) are due to melt production.

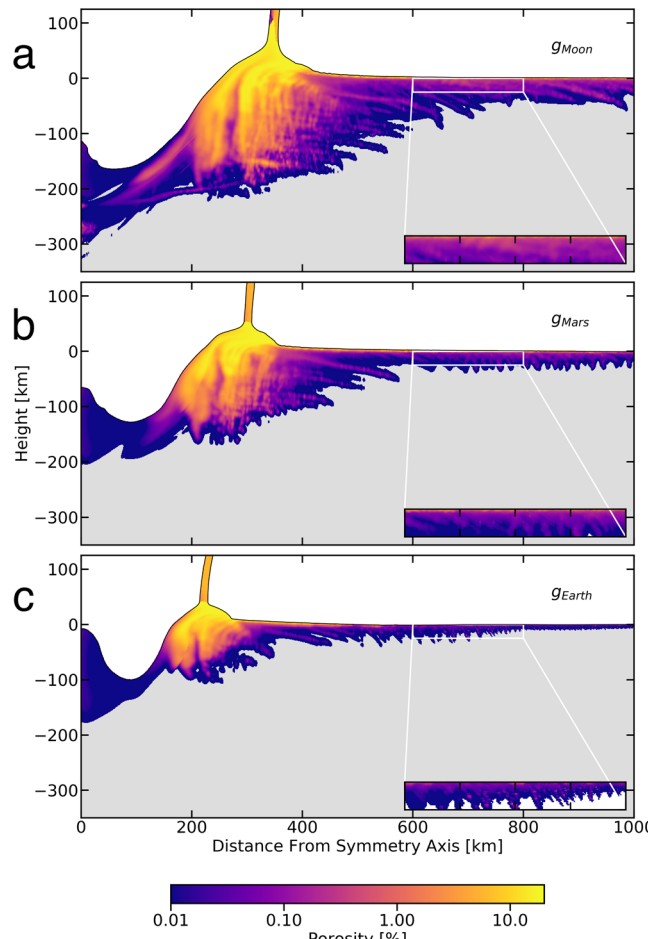

**Fig. 3 | Porosity results for different gravities.** Results of 100 km diameter impactor striking a surface under lunar (**a**), martian (**b**), and Earth (**c**) gravities. The three frames correspond to the color bar with all gray material representing non-porous material. In each panel the zoomed box is the area from 600 to 800 km in the X direction, and 0 to −25 km in the Y direction. Within these zoomed boxes the white material is non-porous material. Space averaged porosity profiles from each of these runs are given in Fig. 4.

gravity appropriate for Mars and the Earth (Fig. 3). Previous work[19] has shown that surface gravity has a substantial effect on tensile fragmentation, and this work confirms that gravity has a corresponding effect on the generation of porosity, with lower surface gravities corresponding to larger regions of tensile fragmentation and porosity. The role of overburden pressure, which scales linearly with surface gravity, is significant, specifically it greatly reduces the depths to which the radially generated porosity is created. Under lunar gravity, at 1000 km radially away from the point of contact, the porosity extends down to ~50 km (Fig. 3a). However, under Mars and Earth gravities the impact-generated porosity only extends down to ~25 km (Fig. 3b) and ~10 km (Fig. 3c), respectively, at a distance of 1000 km radially away from the point of impact. Under Mars gravity our simulated impact-generated porosity on the order of 1% down to a depth of ~3 km beneath the surface ~800 km away from the point of initial contact (Fig. 3b and 4b). In Fig. 4 we present porosity profiles at 400 and 800 km from the symmetry axis taken from our simulations of 100 km diameter impactors striking identical surfaces under different gravities (i.e. the Moon's, Mars', and Earth's). We find that significant porosity is created to relatively large depths at 400 km away on the Moon, Mars, and Earth (Fig. 4a). On the Moon, at ~18 km depth generated porosity is on the order of 1%, on Mars and on Earth there exists several percent

generated porosity several kilometers deep within the crust. Further away from the point of contact at 800 km (Fig. 4a), there exists generated porosities of ~1% at ~3 km deep on the Moon and on Mars. However, at 800 km away from the point of contact on Earth, the porosity generated is much smaller, reaching on a maximum of <0.1% near the surface for the first couple of kilometers of the crust. Additionally, we note that higher overburden pressures create smoother and steeper porosity profiles. It is important to note that this porosity estimate neglects the effect of ejecta falling back down to the surface, which adds an additional layer of porosity in the ejecta blanket. This has important implications for crustal fluid mobility on both early Earth and Mars. Traditionally, impacts are known to generate hydrothermal systems within their rims where porosity is less limited by melts filling void space[25,26]. Our work suggests that impact-generated fluid circulations could have been far more extensive than previously recognized, in regions extending far from the crater rims where melting is negligible.

**Implications for Habitability.** Impact-generated porosity to depths of 3–4 km is conducive to the circulation of near-surface fluids[25–32]. The intense impact flux of the Hadean Earth would have generated widespread hydrothermal systems in proximity of large impacts, with important implication for prebiotic evolution[25–32]. Assuming

the radial extent of impact-generated porosity continues to scale roughly linearly with impactor size (Figs. 1 and 2), we can consider the surface area under which each impact produces porosity in the upper few km of the Earth's crust. Adding the area affected by each Hadean impact we find the cumulative surface area where substantial impact-generated porosity is expected is 1–3 times the surface area of the Earth[31]. In addition, impact-induced fracturing may be even more important in environments more hostile to life, where the fractures could provide a subsurface refuge from variable and harsh surface conditions[32–34]. While the deep lying porosity could dissipate with time, it is important to note that impacts are a protracted source of porosity that could have lasted for 100 s Myr[31,35]. Our work here further highlights the important role of impact cratering in the development and evolution of near-surface habitable conditions on early Earth and Mars.

## Methods

### Description of tensile porosity routine

We use the multimaterial, multirheology iSALE shock physics code to study the role that impacts play in the creation of porosity at depth within the lunar, martian, and terrestrial crusts[18,20–23]. In these simulations we include a dilatancy routine[12], a dynamic fragmentation algorithm[19], and a tensile porosity routine that inserts porosity into material in tension according to simple thermodynamic principles. The dilatancy model accounts for shear-induced pore-space generation that accompanies shear deformation. Due to overburden pressure, the majority of this shear deformation occurs in compression. However, the dilatancy model does not account for porosity that should be created during tensile failure[12]. The dynamic fragmentation algorithm accounts for rate-dependent flaw growth and tensile failure caused by the passage of the shock and release wave[19]. The algorithm reduces the strength of the fractured rock mass and provides estimates of fracture spacing but does not determine the volume of pore space generated. To account for this limitation of previous work, here we employ a simple tensile porosity generation approach.

The full state of stress in a computational cell in iSALE is separated into two parts: the isotropic part and the deviatoric part. The isotropic part of the stress tensor is proportional to the pressure and describes the volumetric and thermodynamic contribution to the stress. The pressure is calculated as a function of density and specific internal energy from the equation of state[20]. The deviatoric stresses, which describe the remainder of the stress tensor, describe the response of the material to distortion. In iSALE, the deviatoric stress tensor is calculated from the deviatoric strain increment tensor under the assumption of linear elasticity and then limited by a shear strength model or envelope[21]. In the shear strength model applied here, shear strength is an increasing function of pressure and a decreasing function of strain. Shear strength vanishes at the intersection of the strength envelope with the pressure axis[21], which defines the minimum or most negative pressure that a material can support, $P_{min}$. The tensile failure model is a supplementary test of the most tensile principal stress, which is reduced if the tensile strength is exceeded[19]. As the equation of state and the strength model are defined independently for a given material, the minimum pressure according to the equation of state will generally not be consistent with $P_{min}$. An additional step is required, therefore, to ensure self-consistency between the two parts of the material model for expanded states.

A portion of the equation of state for basalt[36] used in this work is shown in Fig. 5 for densities close to the zero-pressure reference density $\rho_0$ and a fixed temperature of 300 K. At this temperature, an expanded or distended ($\rho < \rho_0$) state results in a negative pressure in the solid, which represents the interaction energy of atoms located farther apart than their equilibrium positions in the reference state[37].

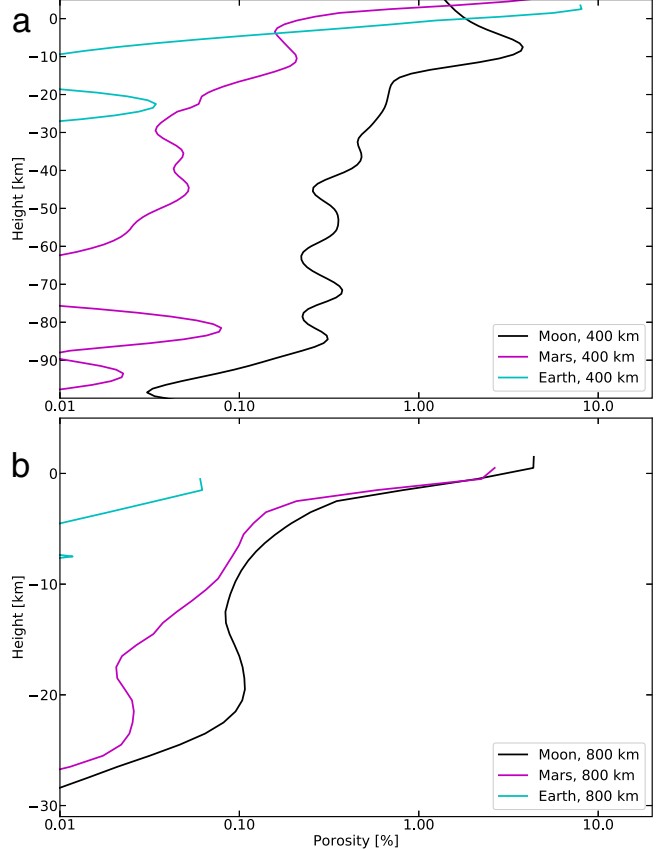

**Fig. 4 | Porosity profiles on Moon, Mars, and Earth after large impacts.** Porosity profiles at 400 (**a**) and 800 (**b**) km away from the point of contact of a 100 km diameter impactor striking surfaces under lunar (black), martian (magenta), and Earth (cyan) gravities. Each profile is horizontally averaged over 11 cells centered around 400 km and 800 km.

For small distensions, the rate of change of pressure with relative density is approximately constant and equivalent to the zero-pressure bulk modulus, implying that relatively small distensions can result in large negative pressures. For a sufficiently distended state (e.g., point A' in Fig. 5), the negative pressure returned by the equation of state can be lower (i.e., greater magnitude) than the minimum pressure that the material is able to withstand according to the strength model ($P_{min}$, Fig. 5), which is a function of damage. In such situations, the standard iSALE solution algorithm caps the pressure in the material at the minimum pressure (point B in Fig. 5), implying that the (capped) pressure and density are inconsistent with the equation of state for the solid. The inconsistency occurs because this approach does not account for the creation of pore space that accompanies fracturing and distension.

The tensile porosity approach implemented here replaces this pressure cap with the insertion of porosity into the cell. In effect, the algorithm prevents the solid material density from dropping below the density that results in a pressure equal to $P_{min}$ (point C in Fig. 5). Instead, porosity is inserted into the cell to achieve the distended bulk density required by mass conservation. In practice, the amount of porosity inserted is determined by iteratively guessing the appropriate density of the solid material, which increases the pressure (relieves the negative pressure) in the material via the equation of state and porosity model[23,38], back to $P_{min}$ (point A'-point C, Fig. 5). This ensures that the pressure and material density remain consistent with both the solid material equation of state and the strength model. The inserted porosity determines the volume of pore space created by distension.

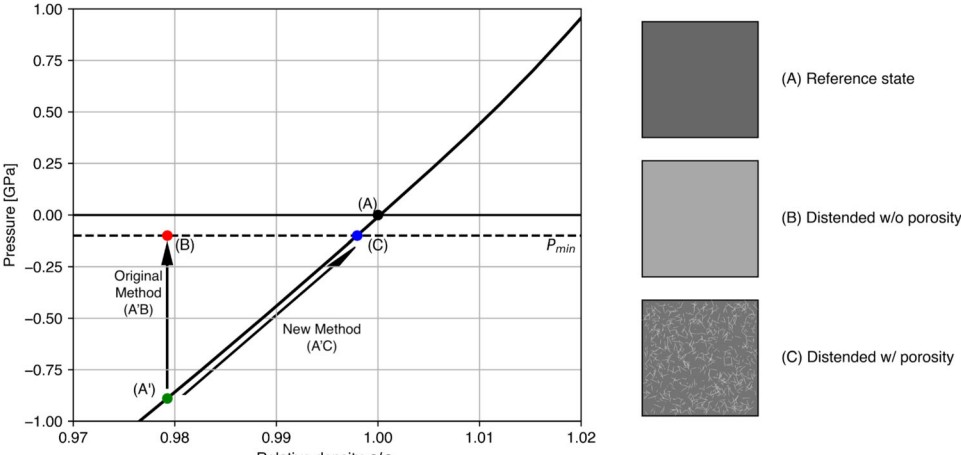

**Fig. 5 | The tensile porosity algorithm.** Material originally at the reference state (**a**; $\rho = \rho_0$) goes into a distended state, $\frac{\rho}{\rho_0} < 1$ (A'), for which the negative pressure predicted by the equation of state (solid black line) for the material is below the minimum pressure permitted by the strength model, $P_{min}$ (dashed line; shown here as 0.1 GPa for illustration). The physical state of such a material is a distended, fractured rock mass. In the original iSALE algorithm, no porosity is added and the solid (and bulk) density is left unchanged, whilst the (negative) pressure is capped at $P_{min}$ (**b**). In the new tensile porosity method, the density of the solid is adjusted to be consistent with the minimum pressure $P_{min}$ (**c**) and porosity (-2% in the example shown) is inserted to fill the remaining volume of the cell and ensure mass conservation.

## Simulation set-up

For this work we simulated vertical impacts into planar lunar surfaces at 15 km/s, a typical lunar impact velocity[39,40]. The impactors we simulated are 1 km, 10 km, and 100 km in diameter, which correspond to final crater diameters of ~21 km, 160 km, and 1200 km, respectively[41]. Both the target and impactor are non-porous basalt. The simulations with 1- and 10-km-diameter impactors are run at a resolution of ten cells per projectile radius, while the simulations with 100-km-diameter impactors are run at 50 cells per projectile radius (or 1 km resolution) to ensure variations of porosity with depth are well resolved. Additionally, to examine the relationship between gravity and the generation of porosity we ran our 100 km simulation under lunar, martian, and Earth surface gravities, all at the same resolution mentioned above.

## Data availability

All data associated with this study are listed in tables and shown in figures. Additionally, all simulation inputs and outputs are available on Harvard Dataverse (https://doi.org/10.7910/DVN/Y09NYD).

## Code availability

Scientists interested in using or developing iSALE should see https://isale-code.github.io/index.html for a description of application requirements. The tensile porosity routines are already available within the latest stable release of iSALE.

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

## Acknowledgements

We gratefully acknowledge the developers of iSALE-2D, the simulation code used in our research, including G. Collins, K. Wünnemann, T. Davison, B. Ivanov, and D. Elbeshausen. We also acknowledge Tom Davison, the developer of the pysaleplot tool, which was used to create many of the figures given in the work. Both S.E.W. and B.C.J. were funded by NASA Lunar Data Analysis Program (80NSSCl7K034l). G.S.C. was funded by UK Science and Technology Facilities Council (ST/S000615/1).

## Author contributions

Firstly, we want to acknowledge the enormous passion and knowledge H.J.M. brought to their life and work, such as this project. His great mind and presence are missed dearly. B.C.J. conceived of the project, G.S.C. developed the basalt equation of state and tensile porosity routine for iSALE, H.J.M. provided direction and insight, S.M. provided their expertize on the consequences of planetary impacts, and S.E.W. collected the data and tied it all together by writing this manuscript with significant writing contributions from all authors.

## Competing interests

The authors declare no competing interests.
