## [Peer Review File · Nature Communications]

REVIEWER COMMENTS

Reviewer #1 (Remarks to the Author):

Gravity data from the GRAIL mission revealed that the crust of the Moon is highly porous (while the upper part of the crust shows higher porosities than deeper regions). Observational data as well as models, simulating impact cratering suggest that porosity is created by impacts, even if the formation of pore space is not fully understood. This manuscript entitled "Widespread Impact-Generated Porosity in Early Planetary Crusts" by Sean Wiggins et al. is raising the question, how much porosity is generated by an impact, in particular with regard to the deeper regions of the crust and outside the crater rim. The authors apply a numerical modelling approach for answering this question. The already existing iSale hydrocode is extended by implementing the effects of tensile fragmentation in a material. The new implementation allows to determine the volume of pore space in the target rock, produced by an impact. By varying the sizes of the impactors and using different gravitational attractions for individual targets, impacts of different strength in different planetary bodies like Moon, Mars and Earth are simulated. The results show that (different to earlier studies) a huge amount of porosity is already created in the first stages of crater formation. Depending on the impactor size as well as on the surface gravity of the target, porosity is created far away from the point of contact in lateral as well as in radial direction. The simulations suggest that for the Moon, impacts creating basins with diameters of 1000 km and more, are capable to produce all observed porosity in the lunar crust, in shallow as well as in deeper regions. For objects with a higher surface gravity like Earth or Mars impacts may be the primary source of porosity as well, even if due to the higher gravitational forces, the emergence of porosity is smaller. The authors conclude that impact generated porosity may have supported fluid circulations in the crust on Earth and Mars which may be very important for the habitability of the terrestrial planets.

Previous numerical models investigating the generation of pore space mainly focus on shear deformation effects (dilatancy). The implementation of tensile fragmentation is a major step forward in completing our understanding of crater formation and the generation of porosity in the target rock. With their study the authors demonstrate, what major significance impact cratering has on the generation of porosity in planetary crusts and explain how porosity is generated also deep below the surface.

The present article is a compelling work and should be published. Below I outline some comments for improving the manuscript that I hope the authors will find useful.

Major comments:

- In my opinion, sometimes a stronger distinction must be made between complex craters and basins. It is useful to speak about craters in general, e.g. when talking about the resulting structure after an impact occurred. But, e.g. in line 24, where it is stated that craters possess high porosities in their centers, this is only true for complex craters, but not for basins (at least

when looking at observational data, when the formation process is completed. Shortly after the first contact, high porosities may also occur for basins, but then I suggest to formulate it more precise).

- Line 129-133: Earlier studies suggest that impacts, which are formed in rock having already a certain amount of porosity, end up with lower porosities. Your results show the same behavior, as discussed in the supplemental material and shown in Figure S2. I would add one or two sentences also in the main article concerning porosity reduction in pre-impact material of high porosity.

- Figure 2: The color bar may be extended with an arrow on the left, showing that values smaller 0.01% are given in grey, same as in Figure 1. I don't understand the exaggeration in vertical direction: The heights given in kilometers are the true values and the distances between single ticks are always the same, or am I wrong? Can you explain the low porosities at a height between 0 km and -200 km in the center of the basin for (B)? The production of melt is not included in the simulations, or am I wrong?

Suggestion for the caption: (A) Resulting porosity 150 s after first contact of a 10 km diameter impactor at 15 km/s. (B) Porosity 500 s after first contact of a 100 km diameter impactor at 15 km/s. Both panels correspond to the color bar at the bottom, given in log scale. Note both (A) and (B) are vertically exaggerated, with aspect ratios of 1.359 and 1.6, respectively. In frame (B) porosity is produced more than 1000 km from the point of contact, outside of the depicted high-resolution zone.

Minor comments:

- Figure 1: There is a white area in the center of the crater for the approach on the left (A), depicting also a region with any or very low porosity? Maybe the time after first contact would be a useful additional information here.

Suggestion for caption: Result of a 1 km diameter impactor striking the lunar surface at 15 km/s. Resulting porosity without new tensile fragmentation routine [X?] seconds after first contact (left). Result including new tensile porosity fragmentation routine [X?] seconds after first contact (right). Both panels correspond to the color bar at the bottom, given in log scale. Grey color represents material that has no porosity or is below 0.01%.

- Figure 3: The color bar may be extended with an arrow on the left, showing that values smaller 0.01% are given in grey, same as in Figure 1 (Another possibility would be to remove the arrow shown in Figure 1, just to be consistent).

- Figure 4: Suggestion for the caption: Porosity profiles at 400 km (A) and 800 km (B) away from the point of contact of a 100 km diameter impactor striking surfaces under lunar (blue), martian

(red), and Earth (green) gravities. Each profile is horizontally averaged over 11 cells, centered around 400 km and 800 km away from the point of contact.

- Line 21: In most publications the "and" in NASA's GRAIL mission is written with small letters.
- Line 26: Instead of "terrestrial" use "Earth" (as done elsewhere in the manuscript).
- Line 26: "scale" may be interpreted as diameter or radius, I suggest to use "diameter" here.
- Line 28: no comma needed here.
- Line 38: small "and" in the name GRAIL
- Line 40-42: The uppermost kilometers of the lunar "highland" crust have an average...
- Line 54-55: There is a typo: "popogates" => "propagates" (2x)
- Line 65: In these simulations we included... should read: In these simulations we "include".
- Line 66: no comma needed before "that".
- Line 93: Either phrase: "into a planar lunar surface" or "into planar lunar surfaces".
- Line 97: ...while the simulations with 100-km-diameter impactors is run at... should read: ...while the simulations with 100-km-diameter impactors "are" run at...
- Line 100: To be consistent, use small letters for "lunar" and "martian".
- Line 168: ...the first couple "of" kilometers...
- Line 179: To be consistent, replace "gray" with "grey".
- Line 203: The minus character was not printed correctly in "iSale-2D".

Minor comments on the supplemental material:

- Line 55: The value of 6.8% is a mean value calculated over different geological units of anorthositic and basaltic rock, or is the average calculated in vertical direction? Please state it more clearly, since the value of 6.8% cannot be found in Besserer et al. (2015).
- Line 65: How will the subsequent porosity generation due to crater collapse modify the porosity (ending up with higher porosity)? And maybe it is also of interest to point out that the crystallization if impact melt in the center will change the picture, too (here, reduction of

porosity)?

Reviewer #3 (Remarks to the Author):

General Comments

This paper treats an interesting subject by attempting to explain relatively new and amazing, high-resolution gravity data with numerical modeling of impacts. While the subject is certainly relevant, I do not believe that the paper is suited for Nature Communications because the journal's format is much too abbreviated. There are simply too many concepts that should be explained or described in considerably more detail to have this paper stand on its own. Simply including citations to do that job does a disservice to the reader, who is then going to be forced to go to a different paper to understand the concept (e.g., the Lennard-Jones cold potential, and others indicated below) clearly enough to follow the rest of this paper. That should be done in the text to an extent sufficient to keep the reader engaged and moving forward. That is lacking here, and doing so would just make the paper too long for Nature Communications.

The shortage of references to previous work is distressing. Dr. Melosh's book is an outstanding piece of work, so much so that I have a copy at work and another at home. I refer to it constantly when I have to review a concept or learn about a new one. Most of it is not, and never posited itself to be, original work, and to cite it instead of the original source bestirs neither a feeling of confidence in the authors nor a sense of respect for that previous work. This shortcoming must be addressed regardless of the journal in which the paper ultimately appears.

The prose of the paper should be tightened up considerably. There are a number of confusing, wordy, or grammatically incorrect sentences that distract the reader from the thrust of the paper.

I'll include one scientific point here in my general comments; other questions and comments appear below. Almost all of the basins were formed early in the history of the solar system, when each of the three targets was hotter at depth than it is today. To make this work more believable, I would encourage the authors to include relevant thermal profiles with depth for the Moon, Mars, and Earth and specifically address how those profiles would affect the rheologies of the supposed target materials. (I'm sure such profiles exist in the literature.) When would the introduction of porosity via fracturing be possible and when wouldn't it? Overburden is one thing, but brittle vs. plastic behavior is another.

Specific Comments

Line 34 — All of the planets in the solar system can be considered as being "ancient." How about "occurring early in planetary history" or something similar?

Line 43 — Croft (1978) [Lunar crater volumes: Interpretation by models of impact cratering and upper crustal structure. PLPSC 9th, 3711-3733] and Croft (1981) [Modification stage of basin formation: Conditions of ring formation. Multi-Ring Basins (P.H. Schultz and R.B. Merrell, eds.), PLPSC 12A, 227-257] should be included here. Indeed, Croft (1981) attempted to explain the porous nature of the lunar crust external to and surrounding lunar basins by impact. Some might not agree with his method in the strictest sense, but *he did it forty years ago*!

Line 48 — This isn't clear. Does "crater rim" refer to the actual rim, or the rim crest? The diameter of the rim is always greater than that of the rim crest.

Line 55 — Reference 14 is absolutely, positively not the appropriate reference to use here. It should be replaced with Shoemaker (1963) [Shoemaker, E. M., 1963 Impact mechanics at Meteor Crater, Arizona. In: B. M. Middlehurst, G. P. Kuiper, (Eds.), The Solar System. IV. The Moon, Meteorites, and Comets. The University of Chicago Press, Chicago, pp. 301-336] and Gault et al. (1968) [Gault D.E. et al. (1968) Impact cratering mechanics and structures. In (B.M. French and N.M. Short (Eds.) Shock Metamorphism of Natural Materials. Mono Book Corp., Baltimore, pp.87-99]. To the best of my knowledge, those were the first explanations, in a planetary-science context, of the mechanism behind the ejection process.

Line 64 — It should be "...within the lunar, martian, and terrestrial crusts," right?

Line 69 — What compression? From normal overburden? From the passing shock front?

Line 75— Starting at this line, the word "pressure" is used a number of times in the rest of the paper where "stress" should be used instead. Such usage doesn't generate a lot of confidence in a paper treating this subject, either.

Line 76 — This isn't worded clearly. Are the material density and specific internal energy the independent variables? If so, it should be reworded, such as "...calculated explicitly using an equation of state with the material density and specific internal energy as independent variables." Otherwise, it should be rewritten accordingly .

Line 78 — There is no such thing as negative pressure. Only stresses are vector (tensor, really, right?) quantities.

Line 79 — The Lennard-Jones cold potential is a critical part of the calculations and should be described, particularly since very few readers will be familiar with it.

Line 81 — The sentence starting at this line really should have a figure to go with it. It would also provide an opportunity to clarify more of these concepts graphically.

Line 83 — The sentence starting on this line is much too cryptic to leave as it is. Not only is the cold potential not described in any detail, but the reason(s) that the "pressure (sic) and density are inconsistent with" it aren't explained at all.

Line 86 — This paragraph is very confusing. As I read it, any increase in porosity in a cell is accompanied by a corresponding increase in density of the material around the pore spaces, and that, in turn, increases the stress in the material. (How and why should be described.) Thus, porosity increases stress in a volume of the material. But the porosity here is a result of fracturing, which is a result of relieving stress, not increasing it. Increasing stress in a volume of material should act to close pore spaces. See my point?

What is the physical mechanism (and by that I mean the real, actual mechanism in the real target, not in the code) that would increase the density of the medium when porosity is created in it? I'm having trouble imagining a process that would increase the density of a material as stress is being relieved in it.

Line 96 — A planar target is almost certainly not a valid condition for an impact of this magnitude, particularly on the Moon.

Line 101 — The word "correlation" here seems to indicate that a result is being assumed. How about "relationship" instead? That's more neutral.

Figure 1 — This figure would be much more informative if the final crater were indicated instead of the growing cavity.

The wording of the caption is awkward. How about this as a possible rewritten version: "Result of the impact of a 1-km diameter projectile into the lunar surface at 15 km/s. The logarithmic color bar on the bottom is applicable to both panels; material below 001% porosity is colored grey. Results without the new tensile-porosity routine are on the left and those including it are on the right."

Line 140 — It's kind of ironic that three full sentences are dedicated to an explanation of overburden pressure, a concept that virtually anyone reading this paper will know and understand, but even a cursory explanation for the esoteric concept of the Lennard-Jones cold potential (and why it's important) can't be found.

Line 152 — Please excuse me for being so direct, but the part of the paragraph starting at this

line and ending at line 159 is bordering on trivial. Anyone who has taken an undergraduate structural-geology course (or anyone who thinks about it for a minute) knows this. We don't need hydrocodes to tell us these things.

Line 169 – The two sentences beginning at this line are confusing. Does "there exists" mean that the porosity's been measured? If not and it's been generated by the modeled impacts, it should be reworded to state that specifically. Should these statements not be dependent on the model results, however, the applicable references should be added.

Line 182 – I suggest changing "extended" to "extensive" in this sentence. "Extended" could be interpreted as having a temporal meaning, which I don't think is the case here.

Figure 3 – How and why does the solid phase transition in the basalt equation of state prevent the creation of porosity? Is that a real effect or one caused by an idiosyncrasy of ANEOS and/or the model? Why does the wedge of unaffected material centered on ~ 5 km depth and a radial distance of 5 and ~ 15 km persist?

Figure 4 – These large impacts occurred early in each planet's history. Without taking the thermal profile of each body into account and how it might affect the rheology of the target with depth, this nice figure has to be considered an academic exercise only.

Reviewer #2 (Remarks to the Author):

Dear Editor and authors,

This paper ("*Widespread Impact-Generated Porosity in Early Planetary Crusts*" by Wiggins et al. [Research Article NCOMMS-21-21860]) focuses on a new routine for the shock physics iSALE code that allows to better understand the identified porosity of the Lunar crust from the GRAIL mission. Porosity of the lunar crust is higher than previously predicted and past numerical models failed to estimate it properly. Authors suggest, for justified reasons, that porosity from tensile strength of a rarefaction wave is key to produce porosity outside of the fragmentation or damage model. They provide with several simulations that gives a good overview of the applicable new routine which, subsequently, leads to results more accurate to the current knowledge of lunar crust porosity.

What makes this journal worth publishing in *Nature Communications* is, on one hand, its numerical addition to a well-known and widely used shock physics code (iSALE). The proposed tensile porosity routine is a great improvement to the code, even in its first step of development. By accepting this paper, the implementation of this routine for the next version of iSALE is assured. On the second hand, the question authors seek to examine is based on recent lunar missions which brought to light important information about the lunar crust. Per my own knowledge, impact studies on the Moon are long standing and it is, for me, of major interest if numerical models address what observations offer us today. Crossing observations and models is key to science and many disciplines would profit from this addition, especially all geoscientists working on the upper lunar crust and future habitable missions.

However, to let this paper being accepted for publication with all the best improvements possible, I find it appropriate to address few moderate comments I will compile in this review. For this purpose, I have used both .docx documents for which I have made comments at specific places in the texts and several text edit suggestions that may help clarifying some concept and/or results I couldn't grasp at first read. These comments and editing suggestions should help authors to really improve their paper and give it more credit and cohesion.

In hope that authors address a vast majority of my comments, I wish to see this paper published, after satisfactory revisions, in *Nature Communications*. If authors want more explanations or do not understand some of my requests, I am happy to help clarify.

With cheerful regards from Tartu, Estonia,

Juulia-Gabrielle Moreau

Research Fellow, shock metamorphism, impacts.

Dept. of Geology, University of Tartu, Estonia

Line-by-line comments from attached .docx files to my review

(the below line numbers reflect the line numbers of the attached files with my text edits, not the original files)

L67-70: Do you mean that pore-space by shear deformation is generated during crater formation and suppressed during compression? I understand this sentence as: "shear-induced pore-space predates compressional phase". Could you maybe make this whole sentence clearer, as I always understood the compression stage is the first to go? Or do you mean that any shear deformation porosity during compression is immediately suppressed by the compression regime itself, at same time?

L93: What would the difference be with non-planar impacts (such as with 100 km projectile), would the curvature have any effect?

L94: Per reading the suppl. material, I guess it is also basalt material for the impactor? What is impactor porosity used, there is no info about it?

L98-99: Is there any test about this resolution choice? It would be interesting to have an idea (as supplementary mat.) and what is the margin of error here :-)

L141: *in order to continue fracturing and fragmenting material:* Maybe you could add: ", which is why no tensile porosity is accounted for beneath the crater of the 100-km-diameter impact scenario (Figure 2B)"

L136-147: General comment on what has been shown so far. Would it be possible to visually separate a bit more between shear/dilatancy porosity with tensile porosity, and see how significant its addition is in term of "apparent area" in the figures? It would be so neat to see that in a Figure, and here, let's say, Fig. 2B :-).

L193-194: (start of section 4) *Impact generated porosity to depths of 3-4 km is conducive to the circulation of near-surface fluids:* Is there some reference for that, or is it drawn from your results, beside the reference in next sentence? I say, because you give a quantitative idea of 3-4 km :-).

L197: I would also refer to Figure 3 here.

L197-199: I am sorry, at first I didn't get the comparison between "cumulative area" and "surface area"? Do you mean that the surfaces affected by impacts, and which were porous from it, has in all history of Earth cumulated an area 1-3 times the surface of Earth? Maybe rephrase that sentence?

L212: Just a little thought here in his memory ♥

L220: (data availability) Will there be some explanatory iSALE modification readme files? Until the next version of iSALE is released, maybe it would be good to tell readers, which use or will use iSALE, that "modifications are as follow"? Or maybe that one author can provide them with the modified routine?

Figure 1: Why not labelling each panel of figure with "no tensile porosity" and "tensile porosity"?

Figure 2: The gray area for <0.01% porosity will blend with high porosities in grayscale. It is not damageable (unlike Fig. 4), but maybe white would be good, or a very very light gray, lighter than the colorbar in grayscale.

Figure 2: To make figure more visible at first glance, I would suggest adding labels in the figures, such as "10 km impactor, t : 150 s" and "100 km impact, t : 500 s", so it really makes these two figures more distinguishable. You could place these in the bottom right corner of each graph in small fonts.

Figure 2: In Fig. 1. you use colorbar that has the <0.1% porosity gray color mark, but it is absent from both Fig. 2 and 3. Could you add it there, as well?

Figure 3, line 182: *nonporous material*: or below 0.1%, yes? Please, specify.

Figure 4: This figure is not colourblind or grayscale friendly, could you change the colours of the profiles to colours readable in grayscale and/or other colourblind categories?

Supplementary material, text S2: Was the tensile porosity routine added here as well? I see in Fig. S2 that the porosity profile beneath the crater is different from Fig. 1 of main manuscript. Could you write the information whether tensile porosity was used, or not? In any case, it would be interesting to cross-check with a simulation that use / or doesn't use the new routine.

Supplementary material, Figure S2: see Figure 4 comment on grayscale/colorblind color scheme

General comments

- In general, the technical application of the tensile porosity is only briefly explained. My background in dealing with iSALE and open data is it would be relevant to have a more detailed explanation on how authors introduced it in the code, how it is affected by equations of state, resolution, extreme cases, heterogeneities, Hugoniot release adiabat, and, if possible, to give instructions to readers and potential iSALE users on how to implement it in their versions of iSALE. I would like, at least as supplementary material, to see some iterations of tests and errors on this new addition and any benchmarking reference if one exists, or particular references to support their results, apart from the one studied in the paper itself. I personally found this lack of technical info a weakness of a well-build paper relaying on such new technique.

- The supplementary material treasures some important info which is not included in the main manuscript. As it is, the manuscript is clear and straightforward, but I'd love seeing the model implied in Fig. S2 to be shown as a final discussion of this paper. Technical details on iSALE parameters can remain as supplementary material.

Reply to Reviewers

Firstly, we would like to thank the reviewers for their thoughtful comments and feedback. We believe that your input has made this a stronger and overall better paper. Secondly, we would like to apologize for the lateness of our response. Unfortunately, one of the authors, SEW, has been in and out of hospitals over the past several months, which has made our reply progress slow. We truly appreciate the patience of not just you, the reviewers, but of the editor as well. Thirdly, we noticed that some of our previous figures were from simulations that used the older basalt equation of state. We now ensure that all figures reflect the use of the updated equation of state. This has resulted in relatively minor differences to the results and figures of this revision (mainly to the 100 km diameter impactors), but we felt it was important to address those differences here. Lastly, we have addressed your comments one-by-one with our corresponding responses in blue font.

Reviewer #1:

Comment 1.1:

“In my opinion, sometimes a stronger distinction must be made between complex craters and basins. It is useful to speak about craters in general, e.g. when talking about the resulting structure after an impact occurred. But, e.g. in line 24, where it is stated that craters possess high porosities in their centers, this is only true for complex craters, but not for basins (at least when looking at observational data, when the formation process is completed. Shortly after the first contact, high porosities may also occur for basins, but then I suggest to formulate it more precise).”

We have changed the line to now read “...with most current models only able to explain high porosity near the lunar surface (first few kilometers) or inside complex craters.”

Comment 1.2:

“Line 129-133: Earlier studies suggest that impacts, which are formed in rock having already a certain amount of porosity, end up with lower porosities. Your results show the same behavior, as discussed in the supplemental material and shown in Figure S2. I would add one or two sentences also in the main article concerning porosity reduction in pre-impact material of high porosity”

In order to clarify and draw attention to previous work we have changed lines 92-97: “While this porosity is modest, our simulations of impacts into an already porous crust agree with previous work¹³ which shows that impacts remove porosity close to the point of impact in an already porous target and will increase porosity further away, especially outside the crater. However, our simulations of impacts into targets with pre-impact porosity indicate that porosity production is additive deep within the lunar crust to yet to be determined threshold (see Supplemental Material).”

Comment 1.3:

“Figure 2:

The color bar may be extended with an arrow on the left, showing that values smaller 0.01% are given in grey, same as in Figure 1. I don't understand the exaggeration in vertical direction: The heights given in kilometers are the true values and the distances between single ticks are always the same, or am I wrong? Can you explain

the low porosities at a height between 0 km and -200 km in the center of the basin for (B)? The production of melt is not included in the simulations, or am I wrong?

Suggestion for the caption: (A) Resulting porosity 150 s after first contact of a 10 km diameter impactor at 15 km/s. (B) Porosity 500 s after first contact of a 100 km diameter impactor at 15 km/s. Both panels correspond to the color bar at the bottom, given in log scale. Note both (A) and (B) are vertically exaggerated, with aspect ratios of 1.359 and 1.6, respectively. In frame (B) porosity is produced more than 1000 km from the point of contact, outside of the depicted high-resolution zone.”

For consistency we have removed the gray arrow for all the figures.

The aspect ratios were chosen in order to present the information in a way such that the two plots (A and B) are easily compared. Additionally, we have addressed the low porosity in the center of the crater by adding this line to the end of the figure caption: *“The low porosity values found within the crater in frame (B) are due to melt production.”*

Comment 1.4:

“Figure 1: There is a white area in the center of the crater for the approach on the left (A), depicting also a region with any or very low porosity? Maybe the time after first contact would be a useful additional information here.”

To clarify what the white colors on the plots correspond to we have added this to the caption of Figure 1: *“White color in the figures in this paper represents void space, or vacuum.”*

Comment 1.5:

“Figure 3: The color bar may be extended with an arrow on the left, showing that values smaller 0.01% are given in grey, same as in Figure 1 (Another possibility would be to remove the arrow shown in Figure 1, just to be consistent).”

Thank you for pointing this out, we have made our figures more consistent now without a grey arrow on the colorbar.

Reviewer #2

Comment 2.1

“Do you mean that pore-space by shear deformation is generated during crater formation and suppressed during compression? I understand this sentence as: "shear-induced pore-space predates compressional phase". Could you maybe make this whole sentence clearer, as I always understood the compression stage is the first to go? Or do you mean that any shear deformation porosity during compression is immediately suppressed by the compression regime itself, at same time?”

This is a good point, as we should have made it more clear in the original manuscript. We have changed the problematic lines: *“The dilatancy model accounts for shear-induced pore-space generation that accompanies shear deformation. Due to overburden pressure, the majority of this shear deformation occurs in compression. However, the dilatancy model does not account for porosity that should be created during tensile failure¹².”*

Comment 2.2

“What would the difference be with non-planar impacts (such as with 100 km projectile), would the curvature have any effect?”

For most of our impacts curvature should not be an important factor. For the 100 km diameter impactor curvature could have a modest effect for the case of the Moon, but likely not the Earth or Mars. We are not currently able to perform simulations including fragmentation with multiple materials. Thus, simulations using central gravity are not possible at this time. We have updated the supplement to explain this to the reader.

Comment 2.3

“Per reading the suppl. material, I guess it is also basalt material for the impactor? What is impactor porosity used, there is no info about it?”

We have updated the manuscript with the following: “Both the target and impactor are non-porous basalt.”

Comment 2.4

“Is there any test about this resolution choice? It would be interesting to have an idea (as supplementary mat.) and what is the margin of error here :-)”

This is a good point. We have added the following clarification to the supplement.

“The resolution chosen for our 1 and 10 km diameter impactor runs was chosen to be 10 cells per projectile radius. This resolution allows us to finish our runs within a reasonable amount of time. For our largest impactor, 100 km in diameter, was run at 50 cells per projectile because we wanted to sufficiently resolve the upper crust. We also ran the 100 km diameter run with a resolution of 10 cpr and the results are similar, but with less variation and fine detail than the 50 cpr runs. This along with the testing of Ref. [19] give us confidence that a resolution of 10 cpr is sufficient.”

Comment 2.5

“in order to continue fracturing and fragmenting material: Maybe you could add: ‘, which is why no tensile porosity is accounted for beneath the crater of the 100-km-diameter impact scenario (Figure 2B)’ ”

We have added the suggested text.

Comment 2.6

“General comment on what has been shown so far. Would it be possible to visually separate a bit more between shear/dilatancy porosity with tensile porosity, and see how significant its addition is in term of "apparent area" in the figures? It would be so neat to see that in a Figure, and here, let's say, Fig. 2B :-).”

Unfortunately, without running more simulations we would not be able to visually separate out the dilatancy and tensile porosity as was done in Figure 1. We believe that this is simply not feasible. However, based upon the work of Collins 2014, the shear porosity is mainly created

during crater modification, so significantly later in time than our simulations consider. Additionally, since most shear porosity is confined to the material closest to the crater, we do not expect significant dilatant bulking far away from the crater rim, such as we see for the tensile porosity in our large simulations.

Comment 2.7

“(start of section 4) Impact generated porosity to depths of 3-4 km is conducive to the circulation of near-surface fluids: Is there some reference for that, or is it drawn from your results, beside the reference in next sentence? I say, because you give a quantitative idea of 3-4 km :-).”

We have updated the references to include the following:

Abramov, O. & Kring, D. A. Numerical modeling of impact-induced hydrothermal activity at the Chicxulub crater. *Meteoritics & Planetary Science* 42, 93–112 (2007).

Osinski, G. R. et al. Impact-generated hydrothermal systems on Earth and Mars. *Icarus* 224, 347–363 (2013).

Baross, J. A. & Hoffman, S. E. Submarine hydrothermal vents and associated gradient environments as sites for the origin and evolution of life. *Origins Life Evol Biosphere* 15, 327–345 (1985).

Corliss, J. B. Hot springs and the origin of life. *Nature* 347, 624–624 (1990).

Deamer, D., Damer, B. & Kompanichenko, V. Hydrothermal Chemistry and the Origin of Cellular Life. *Astrobiology* 19, 1523–1537 (2019).

Lin, L.-H. et al. Long-Term Sustainability of a High-Energy, Low-Diversity Crustal Biome. *Science* 314, 479–482 (2006).

Marchi, S. et al. Widespread mixing and burial of Earth’s Hadean crust by asteroid impacts. *Nature* 511, 578–582 (2014).

Martin, W., Baross, J., Kelley, D. & Russell, M. J. Hydrothermal vents and the origin of life. *Nat Rev Microbiol* 6, 805–814 (2008).

Comment 2.8

“I am sorry, at first I didn’t get the comparison between ‘cumulative area’ and ‘surface area’?”

Do you mean that the surfaces affected by impacts, and which were porous from it, has in all history of Earth cumulated an area 1-3 times the surface of Earth? Maybe rephrase that sentence?”

We have changed the sentence for clarity to “Adding the area affected by each Hadean impact we find the cumulative surface area where substantial impact generated porosity is expected is 1-3 times the surface area of the Earth²⁶.”

Comment 2.9

“(data availability) Will there be some explanatory iSALE modification readme files? Until the next version of iSALE is released, maybe it would be good to tell readers, which use or will use iSALE, that “modifications are as follow”? Or maybe that one author can provide them with the modified routine?”

Currently, the latest version does come with the tensile porosity routine included, and this paper serves as the scientific “readme” if you will. Significant changes have been made throughout the manuscript to reflect this. Otherwise, the data being provided by us, via Harvard’s Dataverse, includes all of our input files (including the latest basalt Equation Of State

file, "basaltg.eos") and should be sufficient for anyone to easily recreate our work even with little experience using iSALE.

Currently, iSALE Dellen contains the tensile porosity used here, meaning it is available on the most recent stable release of iSALE which is freely available for academic use. We have highlighted this by adding a line to the data availability section of the manuscript.

Comments 2.10 – 2.15

These comments specifically address the figures of the paper

Figure 1: Why not labelling each panel of figure with "no tensile porosity" and "tensile porosity"?

Figure 2: The gray area for <0.01% porosity will blend with high porosities in grayscale. It is not damageable (unlike Fig. 4), but maybe white would be good, or a very very light gray, lighter than the colorbar in grayscale.

Figure 2: To make figure more visible at first glance, I would suggest adding labels in the figures, such as "10 km impactor, t: 150 s" and "100 km impact, t: 500 s", so it really makes these two figures more distinguishable. You could place these in the bottom right corner of each graph in small fonts.

Figure 2: In Fig. 1. you use colorbar that has the <0.1% porosity gray color mark, but it is absent from both Fig. 2 and 3. Could you add it there, as well?

Figure 3, line 182: nonporous material: or below 0.1%, yes? Please, specify.

Figure 4: This figure is not colourblind or grayscale friendly, could you change the colours of the profiles to colours readable in grayscale and/or other colourblind categories?

Thank you for these comments. All of these comments have come under consideration, and we have made most of the suggested changes to our figures to reflect the comments made by all reviewers and feel the figures are much improved.

Comment 2.16

“Was the tensile porosity routine added here as well? I see in Fig. S2 that the porosity profile beneath the crater is different from Fig. 1 of main manuscript. Could you write the information whether tensile porosity was used, or not? In any case, it would be interesting to crosscheck with a simulation that use / or doesn't use the new routine.”

The tensile porosity routine is used in all simulations unless otherwise stated (such as in Figure 1 in the main paper). Figure S2 and Figure 1 should not look particularly similar because the simulation shown in Figure 1 does not include any porosity in the target, whereas the simulation for Figure S2 does. The addition of pre-impact porosity will have a large effect on the results due to the difference in energy dissipation. The result shown in Figure S2 can be compared to results of Milburry et al. (2015).

Reviewer #3

Comment 3.1

“This paper treats an interesting subject by attempting to explain relatively new and amazing, high-resolution gravity data with numerical modeling of impacts. While the subject is certainly

relevant, I do not believe that the paper is suited for Nature Communications because the journal's format is much too abbreviated. There are simply too many concepts that should be explained or described in considerably more detail to have this paper stand on its own. Simply including citations to do that job does a disservice to the reader, who is then going to be forced to go to a different paper to understand the concept (e.g., the Lennard-Jones cold potential, and others indicated below) clearly enough to follow the rest of this paper. That should be done in the text to an extent sufficient to keep the reader engaged and moving forward. That is lacking here, and doing so would just make the paper too long for Nature Communications.”

We agree that more explanation was needed. We have significantly expanded our descriptions in response to both reviewers and are still under the length limits of Nature Communications.

“The shortage of references to previous work is distressing. Dr. Melosh's book is an outstanding piece of work, so much so that I have a copy at work and another at home. I refer to it constantly when I have to review a concept or learn about a new one. Most of it is not, and never posited itself to be, original work, and to cite it instead of the original source bestirs neither a feeling of confidence in the authors nor a sense of respect for that previous work. This shortcoming must be addressed regardless of the journal in which the paper ultimately appears.”

We agree and have added significantly to the list of citations throughout the paper.

“The prose of the paper should be tightened up considerably. There are a number of confusing, wordy, or grammatically incorrect sentences that distract the reader from the thrust of the paper.”

In response to both reviewers, we have worked to tighten and clarify the text.

“I'll include one scientific point here in my general comments; other questions and comments appear below. Almost all of the basins were formed early in the history of the solar system, when each of the three targets was hotter at depth than it is today. To make this work more believable, I would encourage the authors to include relevant thermal profiles with depth for the Moon, Mars, and Earth and specifically address how those profiles would affect the rheologies of the supposed target materials. (I'm sure such profiles exist in the literature.) When would the introduction of porosity via fracturing be possible and when wouldn't it? Overburden is one thing, but brittle vs. plastic behavior is another.”

We have run additional simulations in response to this comment and have added our findings and discussion of it into the supplement. In particular we have run our lunar-like target runs for a 100 km impactor into targets with relevant thermal profiles⁴⁸ of 14 K/km and 30 K/km.

Figure S3. Mirrored plot of porosity after 100 km impactors strike lunar-like targets with thermal profiles of 14 K/km (A, on left) and 30 K/km (B, on right). There is little difference between the porosity structures between the two different simulations.

As expected, and now demonstrated by Figure S3 the porosity structures vary only slightly when different thermal gradients are considered. The porosity structures shown in Fig. 3 of the main paper on Earth and Mars are not expected to change significantly either with the addition of thermal gradients mainly due to the fact that in those higher gravity targets result in porosity structures only existing in the upper kilometers of their respective surfaces. Dynamic fragmentation occurs through the growth and linking of cracks/voids at high strain rate and fragment sizes result in a balance of kinetic energy and surface energy. Thus the question of brittle versus ductile deformation is not really relevant. Indeed, the dynamic fragmentation of melts is very similar to that of solids although the surface energies are distinct (Grady, 1982). Thus, our results are relatively unaffected by material temperature. However, post-impact removal of porosity through viscoelastic pore closure is sensitive to material temperature.

We have added the following discussion to the supplement to address the effect of temperature.

“To test the effects of thermal gradients on tensile porosity generation we added realistic thermal gradients of 14 K/km for the Moon⁴⁸ and 30K/km, for our 100 km diameter impactor simulations into non-porous lunar-like surfaces at 15 km/s. The results of this are given in Figure S3 with 14 K/km (Fig. S3a) and 30 K/km (Fig. S3b) showing negligible differences in their respective porosity structure. Though our results show little difference so shortly after impact, it may be important to note that viscous processes and crustal evolution will change these results significantly over time, with steeper thermal gradient bodies viscously relaxing more quickly.”

Comment 3.2

“All of the planets in the solar system can be considered as being ‘ancient.’ How about ‘occurring early in planetary history’ or something similar?”

We have updated the sentence to reflect the reviewer’s suggestion. “Understanding the origin and evolution of planetary crustal porosity is of particular interest because crustal porosity has

a strong effect on thermal, magmatic, and hydrothermal processes occurring early in planetary history¹⁻⁵.”

Comment 3.3

“Croft (1978) [Lunar crater volumes: Interpretation by models of impact cratering and upper crustal structure. PLPSC 9th, 3711-3733] and Croft (1981) [Modification stage of basin formation: Conditions of ring formation. Multi-Ring Basins (P.H. Schultz and R.B. Merrell, eds.), PLPSC 12A, 227-257] should be included here. Indeed, Croft (1981) attempted to explain the porous nature of the lunar crust external to and surrounding lunar basins by impact. Some might not agree with his method in the strictest sense, but *he did it forty years ago!*”

We have added the suggested works to our citations.

Comment 3.4

“This isn't clear. Does "crater rim" refer to the actual rim, or the rim crest? The diameter of the rim is always greater than that of the rim crest.”

Thank you for pointing out this mistake! We have changed our wording throughout the paper to be more specific.

Comment 3.5

“Reference 14 is absolutely, positively not the appropriate reference to use here. It should be replaced with Shoemaker (1963) [Shoemaker, E. M., 1963 Impact mechanics at Meteor Crater, Arizona. In: B. M. Middlehurst, G. P. Kuiper, (Eds.), The Solar System. IV. The Moon, Meteorites, and Comets. The University of Chicago Press, Chicago, pp. 301-336] and Gault et al. (1968) [Gault D.E. et al. (1968) Impact cratering mechanics and structures. In (B.M. French and N.M. Short (Eds.) Shock Metamorphism of Natural Materials. Mono Book Corp., Baltimore, pp.87-99]. To the best of my knowledge, those were the first explanations, in a planetary-science context, of the mechanism behind the ejection process.”

We have updated our manuscript to include the reviewer's suggested references.

Comment 3.6

Line 64 — It should be "...within the lunar, martian, and terrestrial crusts," right?

Correct, we have updated the line accordingly.

Comment 3.7

Line 69 — What compression? From normal overburden? From the passing shock front?

We have updated the line in question to now read: “Due to overburden pressure, the majority of this shear deformation occurs in compression. However, the dilatancy model does not account for porosity that should be created during tensile failure¹².”

Comment 3.8

Starting at this line, the word "pressure" is used a number of times in the rest of the paper where "stress" should be used instead. Such usage doesn't generate a lot of confidence in a paper treating this subject, either.

Pressure (proportional to the hydrostatic or isotropic part of the stress tensor) is the appropriate term. We have clarified the distinction between pressure and deviatoric stress in the context of shock physics modelling. We have also completely revised the description of the

tensile porosity approach to address the next few comments of the Reviewer. We hope that the new description clarifies the approach.

Comment 3.9

This isn't worded clearly. Are the material density and specific internal energy the independent variables? If so, it should be reworded, such as "...calculated explicitly using an equation of state with the material density and specific internal energy as independent variables." Otherwise, it should be rewritten accordingly .

We have reworded this section heavily to improve clarity.

Comment 3.10

There is no such thing as negative pressure. Only stresses are vector (tensor, really, right?) quantities.

Unfortunately, the reviewer is incorrect here. Pressure is a scalar quantity, but negative pressures can exist in solids when they are cold and in an expanded state. This occurs, for example, when the material is pulled in all directions.

Comment 3.11

The Lennard-Jones cold potential is a critical part of the calculations and should be described, particularly since very few readers will be familiar with it.

We have revised our description of the tensile porosity model to describe it in a more general way that is applicable to all solid equations of state and hence have removed reference to the L-J cold potential. The specific form of the cold expanded state part of the equation of state (i.e., the L-J potential in the basalt EoS we use) is not actually that important, because for relevant distensions ($\frac{\rho}{\rho_0}$ less than but close to 1) pressure and density are approximately linear to ensure continuity of $\frac{dP}{d\rho}$ at the reference state, which is why we did not explain it in detail.

Comment 3.12

The sentence starting at this line really should have a figure to go with it. It would also provide an opportunity to clarify more of these concepts graphically.

A figure clarifying the tensile porosity insertion has been added.

Comment 3.13

The sentence starting on this line is much too cryptic to leave as it is. Not only is the cold potential not described in any detail, but the reason(s) that the "pressure (sic) and density are inconsistent with" it aren't explained at all.

The revised description of the approach and an accompanying diagram have hopefully clarified the methodology and the reason for the inconsistency between the state of the material and the solid equation of state introduced by the pressure cap in the standard algorithm.

Comment 3.14

This paragraph is very confusing. As I read it, any increase in porosity in a cell is accompanied by a corresponding increase in density of the material around the pore spaces, and that, in turn, increases the stress in the material. (How and why should be described.) Thus, porosity increases stress in a volume of the material. But the porosity here is a result of fracturing, which is a result of relieving stress, not increasing it. Increasing stress in a volume of material should act to close pore spaces. See my point?

What is the physical mechanism (and by that I mean the real, actual mechanism in the real target, not in the code) that would increase the density of the medium when porosity is created in it? I'm having trouble imagining a process that would increase the density of a material as stress is being relieved in it.

We can understand the confusion. The Reviewer is correct that distension relieves (negative) stress and introduces porosity. This is indeed what happens (A->C, in Fig. 5). To determine this state, however, the material first goes through an intermediate, hypothetical state (A') that assumes no porosity insertion and that the reduction in bulk density is accommodated entirely by expanding the solid component (along the equation of state). Such expansion of the solid leads to large negative pressure. It is the *correction* of this state that requires simultaneously adding porosity and increasing the density of the solid component (which relieves the negative pressure in it). We hope that the revised text and figure resolves the confusion.

Comment 3.15

A planar target is almost certainly not a valid condition for an impact of this magnitude, particularly on the Moon.

For most of our impacts, curvature should not be an important factor. For the 100 km diameter impactor curvature could have a modest effect for the case of the Moon, but likely not the Earth or Mars. Additionally, and most importantly, we are not currently able to perform our simulations on multiple materials, due to limitations on the fragmentation model used, a simulation using central gravity is not really possible at this time. We have updated our supplement to convey this to the reader.

Comment 3.16

The word "correlation" here seems to indicate that a result is being assumed. How about "relationship" instead? That's more neutral.

We have changed "correlation" to "relationship" as suggested.

Comment 3.17

Figure 1 — This figure would be much more informative if the final crater were indicated instead of the growing cavity.

The wording of the caption is awkward. How about this as a possible rewritten version: "Result of the impact of a 1-km diameter projectile into the lunar surface at 15 km/s. The logarithmic color bar on the bottom is applicable to both panels; material below 001% porosity is colored grey. Results without the new tensile-porosity routine are on the left and those including it are on the right."

We have updated the caption of Figure 1, and we believe that is now less confusing thanks to the reviewer's suggestion.

Comment 3.18

It's kind of ironic that three full sentences are dedicated to an explanation of overburden pressure, a concept that virtually anyone reading this paper will know and understand, but even a cursory explanation for the esoteric concept of the Lennard-Jones cold potential (and why it's important) can't be found.

We spent multiple lines within the manuscript explaining overburden pressure because it is central to our results and we want our manuscript to be accessible to readers with a wide range of expertise. Our results show that rarefaction waves can produce such strong negative pressures that they can overcome overburden pressure down to hundreds of kilometers.

Perhaps it is obvious that at some depth the rarefaction waves will no longer be strong enough to overcome overburden pressure. However, we have tried to reword the manuscript throughout to try and drive home the importance of overburden pressure to a central take away.

Comment 3.19

Please excuse me for being so direct, but the part of the paragraph starting at this line and ending at line 159 is bordering on trivial. Anyone who has taken an undergraduate structural-geology course (or anyone who thinks about it for a minute) knows this. We don't need hydrocodes to tell us these things.

We agree that this conclusion that porosity extends to a greater depth on bodies with lower gravity may seem obvious to people who understand geophysics. However, considering the broad readership of Nature Communications and broad expertise among those interested in planetary habitability and evolution of planetary crust, we believe it is important to include any and all backgrounds when drafting publications. Additionally, within the lines noted by the reviewer we report quantitative results from the hydrocode. Although intuition can provide qualitative results or a way to roughly extrapolate our result from one body to another, we do not think a rigorous quantitative estimate of the extent, depth, and magnitude of impact generated porosity can be made without the use of shock physics modeling.

Comment 3.20

The two sentences beginning at this line are confusing. Does "there exists" mean that the porosity's been measured? If not and it's been generated by the modeled impacts, it should be reworded to state that specifically. Should these statements not be dependent on the model results, however, the applicable references should be added.

We have updated the section to now make it clearer that the porosity being mentioned is in fact porosity generated during our simulations.

Comment 3.21

I suggest changing "extended" to "extensive" in this sentence. "Extended" could be interpreted as having a temporal meaning, which I don't think is the case here.

We have made the suggested correction.

Comment 3.22

How and why does the solid phase transition in the basalt equation of state prevent the creation of porosity? Is that a real effect or one caused by an idiosyncrasy of ANEOS and/or the model? Why does the wedge of unaffected material centered on ~5 km depth and a radial distance of 5 and ~15 km persist?

This feature is no longer present in the updated figure that correctly presents results with the updated EOS. Thus, this unrealistic feature was the result of the poorly behaved older basalt EOS.

Comment 3.23

These large impacts occurred early in each planet's history. Without taking the thermal profile of each body into account and how it might affect the rheology of the target with depth, this nice figure has to be considered an academic exercise only.

We believe that we have addressed the reviewer's reservations over lack of thermal gradients previously in this response as well as in our supplementary material. We have found that very different thermal gradients produce strikingly similar results. Additionally, while the reviewer is right that porosity evolution does occur, Figure 4 is more than a mere academic exercise because it provides valuable data to work with, such as starting conditions in nonporous targets.

REVIEWERS' COMMENTS

Reviewer #3 (Remarks to the Author):

The authors have been very conscientious and thorough in their revisions and in addressing my comments, questions, and concerns. That also appears to be the case with the comments, questions, and concerns of the other referees. My only lingering misgiving is the repeated use of "negative pressure." This clearly is not a common term in the physical sciences, and one could argue that it's semantic and idiosyncratic in nature. It could be and, judging from its repeated use in this manuscript, probably is commonly used in the modeling efforts of these (and other?) investigators. Nevertheless, I've asked a number of my colleagues with backgrounds in physical sciences -- including continuum mechanics -- and none of them is comfortable with the term "negative pressure" in this context. I would request, then, that a clear and specific definition of "negative pressure" be given by the authors at its first use in the manuscript. It should use the terminology of standard continuum mechanics or be worded at an even more introductory level. That would allow the diverse readership, cited by the authors in their replies to the reviews, to understand it.

I'd like to thank the authors for taking all of the reviewers' comments seriously and doing such a good job in the revision process!

Mark J. Cintala

Reviewer #2 (Remarks to the Author):

I have reviewed the revisions for the paper "*Widespread Impact-Generated Porosity in Early Planetary Crusts*" by Wiggins et al. [Research Article NCOMMS-21-21860A]) which focuses on a new routine for the shock physics iSALE code that allows to better understand the identified porosity of the Lunar crust from the GRAIL mission.

First of all, I really appreciate authors' replies to all reviewers' comments, and I am grateful for the additional Figure 5 in the manuscript which helps getting our head around the routine. Only in the eye of the beholder, but I may say I really like the new figures with their new color design as per my request, it makes them clearer and friendlier to everyone.

Some requests of Reviewer #3 caught my eyes as well, as of the choice of Nature Communications for the manuscript publication. I think it is now better addressed as well with the new additional Figure.

For this manuscript to be granted my final approval, I look forward to a few applications of my newest comments.

In hope that authors address a fair majority of my new comments, I wish to see this paper published in *Nature Communications*. I do not need to see the revisions if Editorial Team are satisfied by the revisions.

With cheerful regards from Tartu, Estonia,

Juulia-Gabrielle Moreau

Research Fellow, shock metamorphism, impacts.

Dept. of Geology, University of Tartu, Estonia

Line-by-line comments

Note #1: I have made very few editing changes in the attached text document (e.g. impact-generated).

Abstract: As per usual, suggestion time! Would it be relevant to add three words that can change the whole meaning of this abstract? "Using a hydrocode *and new method...*" (tensile porosity)? It's sort of cool to me that the work employed a new or very recent routine in a long-lived iSALE.

Introduction, 1st paragraph: I happily suggest to add this in the paragraph:

"Understanding the origin and evolution of planetary crustal porosity is of particular interest because crustal porosity has a strong effect on thermal, magmatic, and hydrothermal processes occurring early in planetary history¹⁻⁵ *as well as on the prospect of habitable zones outside of Earth.*" (+ any ref.)

L40: "the lunar ~~is~~ crust is much"

L115-127 and Fig. 2: Although this paragraph positively led to understand the role of tensile porosity, and that Fig. 1 is a great example of comparison, it is sadly not really apparent in Fig. 2 and we hang on guesses and wishes to see how strong tensile porosity is. The paper is summarized enough for being accepted without this comment application...

But I'll be stinging of joy if you provide a small snapshot/insert (like in Fig. 3) where the values are not "porosity (%)" anymore, but "approx. tensile porosity gain (%)" (from models without tensile porosity)? Of course, it is unprofessional to "subtract" the *w/o tensile porosity model* (apples) with the *w/ tensile porosity model* (oranges) because the spatial location of materials is different... but maybe it can give a clue on the gain :-)? And if not possible, maybe a numerical clue in the text? Such like "in average, the average gain in porosity deep beneath crater from oldModel to newModel is X%"

L139-141: You can delete this sentence; you mention it later when speaking about Fig. 4.

L141-143: I suggest you change a little bit the sentence and refer way earlier in it that you speak about a new figure :-) I got lost... "In Figure 4 we ..." etc.

L151-152: Maybe rephrase this sentence shorter? -> *Additionally, we note that higher overburden pressure created smoother and steeper porosity profiles.*

On Reviewer #1 comment 1.4 correction and your answer: maybe you don't have to tell anymore that white is void space or vacuum, the empty space they mentioned in the crater is not visible in the new figure anymore :) !

[Comment 1.4:

"Figure 1: There is a white area in the center of the crater for the approach on the left (A), depicting also a region with any or very low porosity? Maybe the time after first contact would be a useful additional information here."

To clarify what the white colors on the plots correspond to we have added this to the caption of Figure 1: "White color in the figures in this paper represents void space, or vacuum."]

Figure 5: It is only a suggestion, but you can either:

- add "old method" and "new method" next to the (B) and (C) labels (e.g. *Distended w/o porosity (old method)* and *Distended w/ porosity (new method)*) ...

- or alternatively/additionally you could do it along the arrows in the graphic; or any suitable labelling...

- or better alternative... Indeed the [A,A',B,C] suite gives the vibe of state of matter and of a **sequence from A to A' to B to C, yes?**... I would prefer A' become B and then B becomes C w/o tensile porosity **and** C' w/ tensile porosity... Or if you pass through C anyway in the routine, then make a D instead of C' and change arrows path accordingly. It doesn't say in the text if you must calculate through the intermediate no-tensile-porosity state to get the tensile-porosity state... So, I suggest one of these two relabeling: **A-B-C-C'** if new method doesn't pass through C... Or **A-B-C-D** if new method does go through C (if you apply this alternative, check in the text and the figure what other things must be changed)

... If you didn't get my point, that's okay, I confuse myself at times :-) !

Supplementary material, text S1 - resolution: Just being picky, but could you previously state that "cpr" is cells per projectile radius -> *cells per projectile radius (cpr)* ?

Supplementary material, Figure S2: Is it possible to use the same colorbar in Fig. S2 as the one used in all other figures? And if not, is it possible to do a visual comparison to a nonporous impact of your main document? If you cannot, that's entirely ok!

Reply to Reviewers

Firstly, we would like to thank the reviewers for their keen eyes, minds and feedback. We believe that your input has made this a stronger and overall better paper, and to express this we have added your contributions to the acknowledgments of our main paper. Lastly, we have addressed your comments one-by-one with our corresponding responses in blue font.

Reviewer #2:

Abstract: As per usual, suggestion time! Would it be relevant to add three words that can change the whole meaning of this abstract? "Using a hydrocode and new method..." (tensile porosity)? It's sort of cool to me that the work employed a new or very recent routine in a long-lived iSALE.

We love the suggestion to 'plug' the novelty of our work, so we have changed a line in the abstract to now read: "*Using new hydrocode routines we simulated fracturing and generation of porosity by large impacts in lunar, martian, and Earth crust.*" Unfortunately, otherwise we are very close to our word limit and without having to sacrifice other points we are pretty limited.

Introduction, 1st paragraph: I happily suggest to add this in the paragraph: "Understanding the origin and evolution of planetary crustal porosity is of particular interest because crustal porosity has a strong effect on thermal, magmatic, and hydrothermal processes occurring early in planetary history¹⁻⁵ *as well as on the prospect of habitable zones outside of Earth.*" (+ any ref.) We have updated the sentence to reflect the reviewer's comment, but we changed the word "zones" to "niches" in order to avoid confusion with habitable zones commonly associated with exoplanets.

L40: "the lunar ~~is~~ crust is much"

We have removed the typo! Thank you for pointing this out!

L115-127 and Fig. 2: Although this paragraph positively led to understand the role of tensile porosity, and that Fig. 1 is a great example of comparison, it is sadly not really apparent in Fig. 2 and we hang on guesses and wishes to see how strong tensile porosity is. The paper is summarized enough for being accepted without this comment application....

But I'll be stinging of joy if you provide a small snapshot/insert (like in Fig. 3) where the values are not "porosity (%)" anymore, but "approx. tensile porosity gain (%)" (from models without tensile porosity)? Of course, it is unprofessional to "subtract" the w/o tensile porosity model (apples) with the w/ tensile porosity model (oranges) because the spatial location of materials is different... but maybe it can give a clue on the gain :-)? And if not possible, maybe a numerical clue in the text? Such like "in average, the average gain in porosity deep beneath crater from oldModel to newModel is X%"

While we would love to do this, it would require us to run more simulations, and without that data we have no way to confidently compare gain of porosity on the non 1 km cases. With that said, however, we have included Figure S4 to our Supplementary Materials, which visually represents the ratio of porosity gained in Figure 1B over Figure 1A, so as to better demonstrate the effect our new routines have. Since tensile porosity increases total porosity by a factor of 10

or more depending on location, we feel this comparison clearly demonstrates that the tensile porosity routine is the dominant source of porosity.

Figure S4. Plot of the ratio of porosity generated with new tensile routines over porosity generated without the new tensile routines, for the otherwise same simulation of a 1 km diameter impactor striking a nonporous lunar-like target at 15 km/s. Material is colored according to the colorbar on the bottom, with all grey material representing nonporous material. It is important to note that this is not a perfect comparison as the two simulation results are inherently different, including spatially.

L139-141: You can delete this sentence; you mention it later when speaking about Fig. 4. While we could delete this sentence, we feel that it fits well as is, and that the information is not repeated elsewhere except graphically.

L141-143: I suggest you change a little bit the sentence and refer way earlier in it that you speak about a new figure :-) I got lost... "In Figure 4 we ..." etc.

We have changed the sentence to be more in-line with the suggestion: "In Figure 4 we present porosity profiles at 400 and 800 km from the symmetry axis taken from our simulations of 100 km diameter impactors striking identical surfaces under different gravities (i.e. the Moon's, Mars', and Earth's)."

L151-152: Maybe rephrase this sentence shorter? -> Additionally, we note that higher overburden pressure created smoother and steeper porosity profiles.

The line has been shortened to “Additionally, we note that higher overburden pressures create smoother and steeper porosity profiles.”

Figure 5: It is only a suggestion, but you can either:

- add "old method" and "new method" next to the (B) and (C) labels (e.g. Distended w/o porosity (old method) and Distended w/ porosity (new method)) ...
- or alternatively/additionally you could do it along the arrows in the graphic; or any suitable labelling...

- or better alternative... Indeed the [A,A',B,C] suite gives the vibe of state of matter and of a sequence from A to A' to B to C, yes?... I would prefer A' become B and then B becomes C w/o tensile porosity and C' w/ tensile porosity... Or if you pass through C anyway in the routine, then make a D instead of C' and change arrows path accordingly. It doesn't say in the text if you must calculate through the intermediate no-tensile-porosity state to get the tensile-porosity state... So, I suggest one of these two relabeling: A-B-C-C' if new method doesn't pass through C... Or A-B-C-D if new method does go through C (if you apply this alternative, check in the text and the figure what other things must be changed)

... If you didn't get my point, that's okay, I confuse myself at times :-) !

We appreciate the suggestions and have updated the figure to more clearly show how the new routine differs from the original.

Supplementary material, text S1 - resolution: Just being picky, but could you previously state that "cpr" is cells per projectile radius -> cells per projectile radius (cpr) ?

Thank you very much for pointing this out to us, we have made it more clear now in the supplement.

Supplementary material, Figure S2: Is it possible to use the same colorbar in Fig. S2 as the one used in all other figures? And if not, is it possible to do a visual comparison to a nonporous impact of your main document? If you cannot, that's entirely ok!

Because we want to highlight changes from zero, we feel a divergent colormap works better in this case.

Reviewer #3:

The authors have been very conscientious and thorough in their revisions and in addressing my comments, questions, and concerns. That also appears to be the case with the comments, questions, and concerns of the other referees. My only lingering misgiving is the repeated use of "negative pressure." This clearly is not a common term in the physical sciences, and one could argue that it's semantic and idiosyncratic in nature. It could be and, judging from its repeated use in this manuscript, probably is commonly used in the modeling efforts of these (and other?) investigators. Nevertheless, I've asked a number of my colleagues with backgrounds in physical sciences -- including continuum mechanics -- and none of them is comfortable with the term "negative pressure" in this context. I would request, then, that a clear and specific definition of "negative pressure" be given by the authors at its first use in the manuscript. It should use the terminology of standard continuum mechanics

or be worded at an even more introductory level. That would allow the diverse readership, cited by the authors in their replies to the reviews, to understand it.

I'd like to thank the authors for taking all of the reviewers' comments seriously and doing such a good job in the revision process!

Perhaps those colleagues working on continuum mechanics were focused on geophysical problems. For many geophysical problems it is true that even if one component of the stress tensor is tensile the mean stress is compressive and pressures are positive. However, in any problem where the mean stress is tensile the pressures are indeed negative. We now clearly define pressure and cite a comprehensive textbook on elasticity in support.

“By convention tensile stresses are positive and pressure is defined as the negative of one third the trace of the stress tensor. Thus, when the mean normal stress is compressive the pressure is positive and when the mean normal stress is tensile the pressure is negative²⁴.”